# Fact and Fiction about 1%: Next Generation Sequencing and the Detection of Minor Drug Resistant Variants in HIV-1 Populations with and without Unique Molecular Identifiers

**DOI:** 10.3390/v12080850

**Published:** 2020-08-04

**Authors:** Shuntai Zhou, Ronald Swanstrom

**Affiliations:** 1Lineberger Comprehensive Cancer Center, University of North Carolina at Chapel Hill, Chapel Hill, NC 27599, USA; risunc@med.unc.edu; 2Department of Biochemistry and Biophysics, University of North Carolina at Chapel Hill, Chapel Hill, NC 27599, USA

**Keywords:** drug resistance mutations, unique molecular identifier, primer ID, next generation sequencing, HIV

## Abstract

Next generation sequencing (NGS) platforms have the ability to generate almost limitless numbers of sequence reads starting with a PCR product. This gives the illusion that it is possible to analyze minor variants in a viral population. However, including a PCR step obscures the sampling depth of the viral population, the key parameter needed to understand the utility of the data set for finding minor variants. Also, these high throughput sequencing platforms are error prone at the level where minor variants are of interest, confounding the interpretation of detected minor variants. A simple strategy has been applied in multiple applications of NGS to solve these problems. Prior to PCR, individual molecules are “tagged” with a unique molecular identifier (UMI) that can be used to establish the actual sample size of viral genomes sequenced after PCR and sequencing. In addition, since PCR generates many copies of each sequence tagged to a specific UMI, a template consensus sequence (TCS) can be created from the many reads of each template, removing virtually all of the method error. From this perspective we examine our own use of a UMI, called Primer ID, in the detection of minor drug resistant variants in HIV-1 populations.

## 1. Introduction

Sanger sequencing has been used for a number of years to look at HIV-1 genotypes and it is the standard-of-care for assessing drug resistance in the viral population in a clinical specimen. Sanger sequencing is interpreted by looking at peaks on a sequencing chromatogram. A single position with peaks from two different bases is considered a mixture of genotypes at that position with both sequences being recorded. There is general agreement that Sanger sequencing is sensitive at detecting such mixtures down to about an 80/20 mix, or a sensitivity of detecting a 20% minor variant [1,2]. Next generation sequencing (NGS) has the capacity to generate almost limitless data in the form of large numbers of sequence reads. This has raised the possibility of detecting minor variants at a frequency much lower than 20%, with many researchers arbitrarily choosing a 1% cut-off as a new limit of detection for minor variants. However, these discussions invariably omit a critical piece of information that is essential for estimating the abundance of variants within the viral population, which is the number of viral genomes that were actually sequenced. The absence of this parameter, and the error-prone nature of NGS platforms, is a recipe for creating data sets that are contaminated with artifacts that form the basis for erroneous conclusions. From this perspective we review the limitations of sequencing HIV-1 populations using NGS and show how the incorporation of a unique molecular identifier (UMI), which we originally called Primer ID [3,4], at the cDNA synthesis step can easily resolve these serious limitations and give high quality and easily interpreted data with a known depth of population sampling. The UMI is a random sequence tag incorporated into the cDNA synthesis primer. After PCR and sequencing, all sequences with the same UMI are recognized as coming from the same original viral genome template. The total number of unique UMIs equals the number of viral genomes actually sequenced (which determines the sensitivity of detection of minor variants), and the pooling of sequences with the same UMI allows the creation of a highly accurate template consensus sequence (TCS) for each individual viral genome that was actually sequenced. Accomplishing both of these steps is essential in order to get NGS data sets that are useful for evaluating low abundance variants.

## 2. The Number of Reads and Sequence Coverage Say Nothing about Sampling Depth of a Viral Population after PCR

“How many pages are in that book?”“Let me look at the last page; there are 500 pages in this book.”“Make 100 copies of 5 pages and put them in a stack; how many pages are in the stack?”“There are 500 pages in the stack.”“How much information is in the stack compared to the book?”“There are 500 pages in the stack.”“I found 1% of M184V in my clinical specimen.”“What is the denominator for that observation?”“I had 100,000 reads with 200X coverage.”

In order to make a claim that a drug resistant variant is present at 1%, that implies that sufficient sampling of a viral population has been done to detect variants that are present at 1% abundance, or 1 in 100 (with 100 being the denominator). If your sample size is 10 viral genomes that were actually sequenced, then it is absurd to claim sampling has been done that could detect a variant at 1%. Similarly, knowing nothing about the sampling depth makes claims of sensitivity nonsensical. However, this scenario plays out every day in the use of NGS to detect minor HIV-1 variants.

PCR is both a blessing and a curse. It is a blessing because it allows us to generate enough nucleic acid mass for downstream manipulation (like loading your favorite sequencer). It is a curse because while we can run the sequencer and get parameters like number of reads and per position coverage, PCR obscures the original sample size of the population that is giving rise to the sequence. There are two powerful artifacts that contaminate deep sequencing data sets. First, because there are so many sequencing reads of the PCR product, the data set consists mostly of reads and rereads of the same small number of original viral genomic templates; for example, it is easy to get 100,000 reads of a PCR product that originally came from 10 templates. The sampling depth of the population is 10 and not 100,000. However, the information of true sampling depth is completely lost if there is no knowledge of the number of templates actually sequenced. This is a phenomenon called PCR resampling and it creates artificial homogeneity in the data set if the output is defined as the number of reads [5]. The second serious artifact is that NGS data sets are the result of a trade-off of volume over accuracy; error rates vary between platforms but can range between 1–10% errors. This creates artificial heterogeneity in the data set. If one compensates for PCR resampling by collapsing to haplotypes (i.e., the number of *different* sequences) then these high error rates create haplotypes that do not really exist, and with some probability that they will appear at positions of known drug resistance markers. Also, collapsing to haplotypes loses information about identical genomes that may have existed in vivo, a different artifactual way of distorting the denominator that defines the sampling depth. 

## 3. The Sampling Depth Is Not the Viral Load

“How many genomes did you sequence?”“The viral load was 10,000 copies per mL.”

Sequencing depth is determined by the number of genomes sequenced (the denominator), not the concentration of viral RNA in the clinical specimen. An aliquot of 1 mL at 10,000 copies of RNA per mL has the same number of RNA genomes as 0.1 mL of an aliquot at 100,000 copies of RNA per mL (10,000 copies of viral RNA). The sampling depth can be no greater than the number of RNA templates that were used at the beginning (and sadly the number that actually get sequenced is much less than the input). Understanding the sampling depth of a viral population requires knowledge that is obscured by multiple steps.
How many RNA templates (i.e., the number) were put into the cDNA reaction?What fraction of the RNA templates were actually copied into cDNA?What fraction of the cDNAs were utilized in the PCR product?What fraction of the PCR product was sampled in the sequencing data set?

If you can account for the efficiency of all of these steps then you can make a true claim that you know what the sampling depth of the viral population was, whether you sequenced 10 or 100 or 1000 genomes, regardless of how many sequencing reads you have. It is relatively easy to get beyond thinking about the concentration of viral RNA in a sample to the number of RNA copies (Volume × Concentration = Mass). However, it is difficult to account for the quality of sampling with the other 3 steps above. Below we discuss the use of a unique molecular identifier (UMI), but as a preview of its utility we will make two points. First, the highest sequencing efficiency we ever see is in the range of 20% of the input RNA templates that actually get sequenced. That means under the best circumstances however many genomes you want to sequence you need to put in five times that number of viral RNA copies at the beginning. Sadly, usually the efficiency of template utilization is even less. Everyone has the experience of having a failed sequencing run. In this case the template utilization was 0. Thus, the efficiency of template utilization (i.e., the fraction of input genomes actually sequenced) varies between 0 and 20%, with 5–10% being common when using clinical specimens. We have also modeled step 4 above. When the viral genomes are sequenced by resampling on average 30 times (i.e., 30× coverage) then more than 90% of the genomes that were sequenced are represented in the data set of total reads [6]. This is a useful parameter when deciding how many specimen libraries to pool in a deep sequencing run to allow a sufficient number of reads for each specimen.

## 4. The Number of Genomes Actually Sequenced (the Denominator) Determines the Sensitivity of Detection

“I didn’t detect any variants in that clinical specimen you gave me at 1% sensitivity.”“How many genomes did you sequence to get that level of sensitivity?”“I used 100 μL.”

Above we gave the example of the absurdity of claiming detection of a variant present at 1% if only 10 viral RNA genomes were actually sequenced (even though there were 100,000 reads and 200× coverage). This gets at the issue of the sensitivity of detection as a function of sample size. Here of course the sample size is the number of viral genomes actually sequenced, which forms the denominator for the value of frequency (which times 100 gives the % abundance). There is a simple way to approximately estimate the sensitivity of sampling with the goal of saying there was a 95% chance of detection something that was present at X% abundance. If N is the sample size (here the number of genomes actually sequenced) then:

Detection limit or detection sensitivity can be calculated based on upper 95% confidence limit for the Binomial proportion when no event has been observed. In the R program we can use *binom.test*(*0*,*N*) to calculate the confidence interval using Clopper–Pearson method and use the upper limit as the detection sensitivity for N number of sequences. For a quick assessment of data, we can use 300/N to estimate the sampling sensitivity (in %) to detect a variant with 95% confidence. 

Let us go back to our desire to claim 1% sensitivity. Using this formula, it is clear that a sample size of 300 genomes sequenced is necessary to have 95% confidence that a variant present at a true abundance of 1% is detected; on average you should detect it 3 times with a sample size of 300 but because sampling is random you may detect it 1, 2, 3, 4, 5, etc., times but with a 95% chance of detecting it at least once. A separate question is the accuracy of the estimate of abundance and this is determined by a different parameter. If you detect something one time you do not have much information about its true abundance. To estimate that value, it is important to detect a variant multiple times to be able to put a confidence interval on its detected abundance. In this case if you had a sample size of 300 viral genomes sequenced and you detected a variant 3 times you could report its presence at 1% with a confidence interval of between 0.2% and 2.9% (Exact Binomial Test). If you wanted to improve the accuracy of estimating true abundance when trying to detect a variant at 1% then it is necessary to increase the number of observations by increasing the sample size; for example, sequencing 3000 genomes and detecting the variant 30 times allows you to report a variant at 1% abundance with a confidence interval of between 0.7% and 1.4%. We will remind you that every day, people are using NGS to make claims of 1% sensitivity [7,8] without any knowledge of the number of genomes sequenced. 

## 5. How to Fool Yourself into Thinking Your NGS Protocol Gives You 1% Sensitivity

“I climbed Mt. Everest, it’s that hill over there.”“Really? It doesn’t look very high.”“Yes, but there is a sign on top that says ‘Mt. Everest’.”

It is easy to get any NGS run to document the presence of a variant at 1% abundance and with very high accuracy. However, this is about the same as sticking a sign on a nearby hill that says “Mt. Everest” and then claiming you climbed it—it is not a real-world situation.

The simple way to get a great estimate of abundance is to mix two samples at high viral RNA concentrations and then sequence many genomes without even having to validate the number of genomes sequenced. In the extreme one can mix PCR products or tissue culture supernatants containing two viruses, one being at 1% abundance. There will be lots of independent copies of each variant that make it into the sequencer and it will give very accurate abundance [9]. You have climbed Mt. Everest with your NGS protocol proving you have the ability to detect variants at 1% abundance. However, if you do the thought experiment of diluting the input of mixed RNA templates, at some point your ability to measure the 1% variant will diminish. Using the Poisson distribution, you can calculate that if you sequenced 100 genomes, 37% of the time you would fail to see the 1% variant, and if you sequenced just 10 genomes then 90% of the time you would fail to detect the variant present at 1%. Worse yet, you might record a sequencing error as being present at 1%. Welcome to the real-world Mt. Everest.

Clinical specimens are challenging because the amount of material is limiting (i.e., the number of RNA copies that actually go into the sequencing reaction), and the efficiency of sequencing the RNA genomes varies between specimens and can be poor. Just because your protocol CAN detect 1% abundance does not mean it DID detect 1% abundance with the last clinical specimen tested. Let us take a typical real-world scenario. You get 100 μL of a sample that has 30,000 copies of HIV-1 RNA per mL. “Volume X Concentration” tells you that you have 3000 copies of RNA. In your protocol you extract the 100 μL and put half into a cDNA reaction (1500 copies of RNA). Without knowing the efficiency of cDNA synthesis you then use only half of the cDNA product in the PCR (so a max of 750 possible copies). In reality your efficiency was not bad at 10% template utilization so you wind up sequencing 75 original RNA templates. With this level of sampling you have 95% confidence of detecting a variant at 4%. The next sample is exactly the same but the starting viral load is 15,000 copies per ml, so your sensitivity of detection is 8% (with 95% confidence). You previously validated that your method could detect variants at 1% with high accuracy using virus-containing tissue culture variants. Do you report an absence of variants at 1% even though your true sampling sensitivity varied between 4–8% sensitivity? That would be bad form. Equally bad is not to know the sensitivity (i.e., the sampling depth) and report your results as if you had 1% sensitivity. You can go through this exercise but cut the viral load by 10 (“looking for low abundance variants in samples with low viral load”) and see again how this drifts into the realm of the absurd.

## 6. Population Skewing during PCR with Low Template Number

“Of all the people who started the marathon only the skinny people finished in under 3 h.”

If you use NGS of HIV-1 with clinical samples you probably have to admit to yourself that (i) you do not know how many templates you actually sequenced, and (ii) you may have sequenced only a small number of viral RNA genomes in at least a fraction of those samples. Using a UMI it is possible to follow the fate of each cDNA molecule through PCR (the details of the introduction of the UMI are described below). For this discussion just imagine you can watch the fate of each copy of viral RNA that got copied into cDNA during the PCR. Each cDNA has a unique UMI but since PCR amplifies each cDNA that UMI (and its attached cDNA) now appear as multiple copies and thus get sequenced multiple times. However, not all cDNAs experience the PCR exactly the same, some get amplified more, some less, giving rise to a distribution of the number of times each original cDNA is represented in the final sequence data set, and as you sample the PCR product as molecules that got sequenced, two variants present at the same concentration may not get sequenced an equal number of times. In addition to this distribution there are two other features of the data set worth noting.

Figure 1 shows a typical distribution of unique UMIs (Primer IDs) by their frequencies in the sequence reads. Without bias we would expect the distribution of UMI frequencies to fit a Gaussian distribution simply due to template sampling of the PCR product during sequencing (simulated as the red curve in the figure). However, the actual distribution of Primer IDs is wider than the simulated one, suggesting that PCR skewing is also impacting the distribution of the frequencies of representation of the original cDNA templates. In this example, two different cDNAs can vary by 10-fold in abundance in the sequencing data because of the PCR skewing and PCR product sampling. Thus, if there is one copy of an interesting sequence variant among the genomes that you actually sampled, its detected abundance in the sequenced PCR product can be skewed 3-fold higher or 3-fold lower. We have been able to approximate this outcome with a simple model. The model assumes that PCR is only 50% efficient at including a template in each cycle of DNA synthesis. At the starting template number of input cDNAs, only half are copied during the first PCR cycle meaning they are now at twice the copy number as those not used as template. If this happens at each round you can see how some sequences by chance become over-represented in the first few rounds of PCR while some by chance do not get utilized in the first few round and become significantly under-represented. As the copy number of each original template increases during PCR these differences average out, making just the first few rounds problematic. Below we discuss how the use of a UMI corrects for this problem.

However, the use of a UMI introduces two types of artifacts in the sequencing data set that need to be understood but that are easily dealt with. First, the error-prone sequencing platforms (and PCR mis-incorporations) can introduce sequence errors into the UMIs themselves. Remember that we track a single cDNA by its UMI and when the same UMI is seen multiple times in the PCR product we can ascribe those reads as having come from the same original viral RNA template. Thus, when some of these identical copies have a sequencing error in the UMI this makes it look as if it came from a different cDNA (we have called these offspring). Fortunately, these errors are individually present at low abundance, thus the true UMIs (tagging the original cDNAs) are present at much higher abundance than the offspring UMIs in the data set. These offspring UMIs are seen at the low read number but high total number of “unique” UMIs (green dots, left side of the distribution) in Figure 1. We have developed an algorithm using the platform sequencing error rate and the sequence coverage depth to filter out the offspring UMIs (and their attached sequences) from the sequence data set to remove these artifacts [4]. We simulated the highest possible frequency of the offspring UMI/PID given the highest frequency of actual UMI and the raw sequencing error profile, and use this as a cut-off for artifactual UMIs (blue dotted vertical line on Figure 1). However, using this cut-off we may sacrifice some real UMIs as we can see the parent UMI distribution is overlapping with the offspring UMI distribution (through a simulation) on Figure 1. The latest TCS pipeline gives the users more options to adjust the cut-off value to include/exclude more UMI/PID around the offspring cut-offs. 

The second bias we observe is that a few UMIs have very high read numbers (orange dots in the figure at the right of the distribution). We believe these are present for one of two reasons (not mutually exclusive) and the use of a UMI solves both potential problems. These could be extreme examples of PCR skewing; since the use of a UMI allows all sequences with the same UMI to be collapsed it does not matter if there are more or less of any one UMI, it is still counted just once. Alternatively, when the UMIs are incorporated it is possible by chance for two different cDNAs to be synthesized with the same attached UMI; in this case the high read number is really the sum of two different cDNAs. However, one of the cDNAs will be represented with a higher number of sequences reads (by chance, given there is a distribution) and the less abundantly read cDNA will just be lost when sequences are collapsed to create a consensus sequence for that UMI. Since deep sequencing should be used to sequence dozens to thousands of RNA genomes, the loss of a few template sequences in the data curation is not a significant problem.

## 7. Sources of Sequencing Error in NGS Protocols

“Look at all the snowflakes, it’s all so white and homogeneous.”“Yes, but if you look closely each snowflake is different.”

Sanger sequencing remains the gold standard sequencing method in HIV DR clinical testing because of its high accuracy, up to 99.99% per specimen [10]. In reality this is an “apparent” accuracy since Sanger sequencing is generating a single consensus sequence of a bulk PCR product; minor errors in the misincorporation of the wrong chain terminating nucleotide are hidden in the bulk of correct incorporation such that a 1% error rate would go completely unnoticed. Unlike Sanger sequencing, NGS is error-prone as sequencing devolves down to the single template level. The sources of sequencing errors in NGS protocols can be from misincorporation by reverse transcriptase at the very first cDNA synthesis step (about 1 in 10,000 nucleotides), nucleotide misincorporation, the sequencing platforms themselves, and bioinformatics/correction pipelines. The method error rate at each final nucleotide position can be as high as 1–10%, depending on the combination of library protocol, sequencing platform, and bioinformatics tools. If NGS is used to generate a consensus sequence for a sample or even to look for variants at 20% abundance, then it is likely to be comparable to the accuracy attained using Sanger sequencing (we discuss why this might be below). However, the high method error rate when trying to sequence individual viral genomes to sample the population raises huge concerns about the accuracy of calling minority variants. 

Another type of errors is from the template switching and recombination during PCR. This type of error may not be noticed by some of the HIV NGS DR protocols that cannot obtain mutation linkages, also known as mutation linkages lost at the library prep step through fragmentation. But it is particularly important for those who study the correlations between different mutation sites. 

In an effort to demonstrate the background noise of a typical HIV-1 NGS DR assay and its impact on the appearance of minority variants, we analyzed two specimens as raw sequence reads and then as template consensus sequence (TCS) reads using a UMI (Primer ID), with each viral RNA amplified and sequenced independently three times. The raw sequences went through the same bioinformatics pipeline to filter out low-quality reads (i.e., APOBEC3G/F hypermutations, frameshifts, and stop codons) as the TCS reads. Figure 2. shows selected DRM positions in the RT region of the raw and TCS reads. Panel (a) shows the data from viral RNA extracted from particles produced from the 8E5 cell line [11,12]. 8E5 cells are clonal with a single copy of a defective proviral DNA per cell, thus the genomes in the particles produced by the 8E5 cells are genetically homogenous, making it an ideal control for HIV-1 NGS DR assay. Using the raw sequence reads we detected “apparent” DRMs across the sequenced HIV RT region in the range of 0.1–1%, with the M184I mutation breaking through the 1% cut-off in the three replicates, which would have been reported as a minority DRM by other protocols. However, in the TCS reads we did not detect any DRMs in this homogeneous viral population. Since the use of a UMI allows an assessment of the number of templates actually sequenced we could tell that we had an average detection sensitivity of 0.47% in the three different sequencing runs. Panel (b) shows the comparison of raw sequence reads (top) with UMI/Primer ID TCS reads (bottom) from a clinical sample we have previously described [13]. In all the three replicates, we observed the L210W mutation in the raw reads with a frequency around 10% and agreeing well among the replicates. In addition, we also saw several other DRMs above the 1% cut-off in the raw reads. In the first replicate the raw sequence reads contained K65R at 2.1%. The same replicate also had 1.2% Y181C and Y181I, and in different replicates V179F reached and F77L approached the 1% cutoff. Comparison with the UMI/Primer ID TCS analysis *of the same data* reveals several important points. First, we can see that the number of unique templates sequenced varied between 67 and 97. This level of template sampling (present in the raw reads but only determined by the UMI) explains why for these samples there is good agreement for the variant present in the 10% abundance range (L210W). The detection of K65R at 2.1% abundance appears to be an artifact of analyzing the raw reads. It is not present in the UMI TCS analysis; however, because of the limited number of TCS per replicate (67–97) there is limited sensitivity to detect a variant at 2%. Another virtue of using a UMI is that since the replicates are also independent samplings of the viral population they can be pooled to increase the sensitivity (81 + 67 + 97 = 239; 1.5% sensitivity of detection with 95% confidence), further suggesting the detection of K65R is an artifact. Y181C and Y181I were detected in at least one replicate in each analysis suggesting they are really present but at such low abundance that their detection fluctuates in the different samplings, somsething that can be inferred only once the number of templates sequenced is known. Finally, the detection of V179F and F77L appear to be artifacts in the raw reads since even a single detection linked to a UMI would have been recorded in the TCS analysis (remember, these are the same data). This points to the very significant problem that in the raw reads the detection of Y181C and Y181I may be correct but the inclusion of K65R, F77L, and V179F would inflate the detection of minor variants without being able to distinguish which are true and which are false. From this analysis it should be clear that the use of a UMI does not improve the depth of viral genome sampling but rather reveals that depth of sampling allowing the data to be analyzed with appropriate rigor and limits. Also, these were the errors we observed in this batch of raw sequences, this does not define the errors that will be present in the next batch of raw sequences. A better strategy is to remove the errors with the use of a UMI.

There is one other point worth making in this type of analysis. As noted above, there is an RT error rate during cDNA synthesis of about 1 nucleotide in 10,000 bases. Since these errors are incorporated at the time the sequence is linked to the UMI these cannot be corrected by downstream manipulations. This appears to be the residual error rate we can detect using the UMI/TCS approach as defined by sequencing the viral RNA produced from 8E5 cells (this number also includes RNA pol II error rates during RNA synthesis). This error rate can be included in the analysis of the TCS. For example, if the amplicon being sequenced is 500 bp then every 20 TCS will have 1 artifactual nucleotide error. In replicate 1 of the clinical specimen shown in Figure 2b there are 81 TCS and thus about 4 random errors in the data set. It is possible to assess the frequency of a low abundance variant relative to the residual method errors that must be present to provide a likelihood that the observed variant is potentially the result of residual method error [6]. This type of analysis significantly adds to the rigor of assessing the true presence of low abundance variants.

## 8. How Does a Unique Molecular Identifier Fix All of These Problems (It Seems Like Magic)?

“All these cars look the same. How do you know which is yours?”“I look at the license plate.”

UMI is a simple and straightforward solution for all the issues of conventional NGS in HIV-1 DR testing that we described above. During the cDNA synthesis step, a cDNA primer is used with a degenerate block of nucleotides to tag each RNA template with a unique ID (the Unique Molecular Identifier or Primer ID) (Figure 3a,b). During the cDNA primer synthesis, a stretch of consecutive nucleotides is randomized to make a library of cDNA primers that are then randomly sampled to initiate cDNA synthesis. The “birthday paradox” (in a room of 50 people you have a 95% chance of sampling a birthday twice), i.e., the chance resampling of the same UMI attaching to different RNA templates, can be minimized by using a large excess in the number of possible sequence combinations of UMIs relative to the number of input RNA templates. The number of possible combinations of UMI is determined by the length of UMI. A degenerate block of 10 bases can generate UMIs of 4^10^ (1,048,576) combinations. We normally use no more than 1:100 ratio of input number of templates to UMI combinations to minimize resampling of the UMI. In clinical HIV-1 DR testing, a UMI of 11 or 12 bases long sufficient to serve this purpose. An alternative way of adding the UMI is to ligate a degenerate oligonucleotide after cDNA synthesis [14]. After the UMI tagging, PCR amplification is used to amplify the cDNA just like other library prep protocols. The UMI tags are included in the amplicon during all of the downstream processes. After sequencing, the UMIs will be first tabulated by their frequencies and processed to remove the offspring UMIs (green dots in Figure 1). As discussed above, the over-represented UMIs due to the PCR skewing/UMI resampling (orange dots on Figure 1) can be retained. Sequences with the same UMI will be collapsed to make a template consensus sequence (TCS) to represent the original viral RNA template. The TCS retains any errors made by RT, but sequencing errors introduced during PCR or during the sequencing itself, while still present in the data set, are all lost during the creation of a consensus sequence for each template based on the multiple reads obtained by PCR resampling. This process will also eliminate recombinants created during PCR, and reveal the mutation linkages on the same template. Recombination during PCR occurs when there is incomplete synthesis of a strand during one round and that incomplete strand anneals to another strand and gets extended during a subsequent round (template switching); these recombination events typically occur later in the PCR when the product can compete with the PCR primers for annealing to the single strands, thus individual recombinants are of low abundance and thus lost during formation of a template consensus sequence. Usually a consensus sequence is made by majority rule, but this step can be made even more rigorous using a super-majority rule to define usable TCSs [14]. The total number of different UMIs (each linked to a TCS) is the sampling depth of that particular specimen at the sequenced region. Thus, we have the right denominator (sample size) to calculate low level DRMs correctly. Knowing the sampling depth, we are also able to calculate the detection sensitivity and the confidence intervals of the abundance of the detected DRMs. These are all essential parameters needed to truly define the presence of minor variants.

As we can see, the inclusion of a UMI for HIV-1 DR testing is not much harder than a typical NGS protocol, with the inclusion of a UMI tagged sequencing primer and an additional purification step to remove the unused cDNA primer so UMI tags do not get introduced during the PCR. Appendix A shows a brief outline of the steps we use in the Primer ID protocol. Most of the steps will be very familiar to those using NGS for HIV-1 DR testing.

The inclusion of a UMI in the cDNA primer does require extra data processing (but not much). In Appendix A is a flow chart of the data processing pipeline using data taken from the MiSeq platform and with pooling of specimen libraries using Illumina indexes. We have created a largely automated version of this pipeline with a web-based interface to allow anyone to analyze their Primer ID data in the context of sequencing HIV-1. Access is freely available (and encouraged) through this link (https://tcs-dr-dept-tcs.cloudapps.unc.edu/).

Our use of the Primer ID/UMI protocol has been limited to the MiSeq platform. The current standard use of this platform is to generate a long cDNA and amplicon spanning the regions of interest (the coding domains for protease, RT, and integrase). That single amplicon product is then fragmented and sequenced as fragments with “coverage” used to estimate sampling depth and with loss of linkage between any two variants. It should be immediately obvious that there is no chance for knowing how many templates were used to generate the long PCR product, although the long amplicon and fragmentation allow use of the MiSeq platform which has a maximum sequence read length of about 600 nucleotides using the 300 base paired-end read protocol. To overcome the read length limitation for this platform in the context of HIV-1 drug resistance testing, we have developed a multiplex protocol where 4 or 5 different cDNA primers (with Primer ID) are included in a single cDNA reaction, the multiple cDNA products are amplified (500 bp each) with a common downstream primer (encoded at the 5′ end of the cDNA primer) and 4 or 5 gene-specific upstream primers. Using this approach along with Illumina indexes (which identify the individual clinical specimens), we are able to generate deep sequencing data for minor drug resistant variants across protease, RT, and integrase for up to 24 specimens in a single MiSeq run. The sequence output is first sorted by the Illumina index (i.e., each clinical specimen), then by amplicon region, then by UMI to generate TCS for analysis of drug resistant mutations. In this approach the detected mutations within an amplicon can be interpreted for linkage but those between amplicons cannot [15,16]. However, the multiplexing protocol may lead to a significant loss of sequencing reads of targeted region if the number of multiplexed regions and specimens are not calculated properly. We normally require an average number of 30 raw sequence reads per valid Primer ID. Given an input RNA template number of 20,000 and a template utilization of 10% (i.e., 2000 RNA templates actually get sequenced as measured by the presence of independent PIDs), to have 4 regions sequenced, we need 240,000 raw reads. We usually see 40–50% of raw reads either have low quality or will be discarded as “offspring” sequences. Thus, we need at least 400,000 raw sequence reads per specimen. A normal MiSeq run (mixed with 15% PhiX to increase quality) will return around 10 million raw sequence reads, and if we pool 24 samples for each MiSeq run, the raw sequence read numbers should be enough to recover the desired number of TCSs (in this case 2000 per region) for each region and for each specimens. The number of specimens pooled can be adjusted based on the estimated number of input templates, but our format of pooling 24 specimens each with 4 multiplexed regions should work for most of the HIV clinical samples. The choice of an average of 30 reads per TCS is made to ensure that most of the templates that were sequenced will appear in the TCS data set, i.e., all sequenced templates will be sampled in the raw reads when making TCSs.

## 9. Why 20% Abundance Is the Useful Claim for NGS Sensitivity (and Why Sanger Sequencing Works at This Level)

“It doesn’t matter if I grab one handful or two handfuls of these jelly beans, there is always about 20% of the red ones.”

We have put a great deal of emphasis on sampling depth of the viral genome population and specifically the need to know that depth to validate inferences made about minor variants. Sampling depth comes from the number of viral genomes actually sequenced. There are protocols where the user intentionally generates sequence from a single copy of the viral genome (end-point dilution PCR/SGS/SGA) making the sample size of the viral genome population equal to the number of independent amplicons sequenced. However, most PCRs are not designed to be sensitive to giving a robust product from a single starting template. Thus, one can imagine that a typical successful PCR that generates a detectable product (enough to sequence by Sanger sequencing) had to start with some minimal number of templates. To have a 95% chance of detecting a variant present at 20% one needs at least 15 sequenced genomes (300/15 = 20% sensitivity of detection). This is for detection as there would be 3 copies on average of the variant present at 20% in the 15 copies sequenced, making its measurement sensitive to sampling and PCR skewing. To improve accuracy of estimating abundance at 20% lets up the number of templates sequenced (not the number of RNA templates put in the cDNA reaction but the number actually sequenced) to 50 (this would also increase the detection sensitivity to 7%), a number that is usually reached with an input of 500–1000 copies of viral RNA into the cDNA reaction for most specimens. Thus, protocols that start with at least 1000 copies of viral RNA (with all of the cDNA product going into the PCR) will usually get 20% accuracy of detection, either by sequencing the bulk PCR product using Sanger sequencing and looking for mixed peaks or by reading the raw reads in the NGS output (although the depth of sampling will not be validated). The idea here is that with an input of 1000 RNA copies into the cDNA reaction (and using all of the cDNA product in the PCR), a poor template utilization of between 1.5% (i.e., 15 copies) to 5% (50 copies) will give sampling sensitivity of minor variants in the range of 7–20% abundance within the PCR product (assuming no skewing during PCR). However, with the higher number of templates sequenced (i.e., 50) the confidence interval for detecting something at 20% is 10–34%, significantly better than if something is sampled at 20% with the smaller number of templates (15), which has a wide confidence interval of 4.3–48%.

There are several points to make here. First, in the absence of using a UMI we should not pretend we have more sampling depth than we can validate. Using the fortuitous efficiency of a typical PCR as the bar argues for using the same 20% cutoff for NGS data as has been used for Sanger sequencing—unless a UMI is used. Second, at the recent WHO Workshop in Winnipeg, Richard Harrigan argued that most of what we need to know about HIV-1 drug resistance in the clinical setting can be learned based on 20% sensitivity of detection, suggesting either Sanger sequencing as it is currently used or NGS without a UMI and limited to a 20% cutoff could be used going forward in patient care. Third, if we want to explore the significance of minor variants present at less than 20% abundance (whether it is 10% or 1% or 0.1%) it will be important to incorporate a UMI strategy to allow the distinction of true variants from contaminating artifacts in the sequence data sets, and to define the true depth of population sampling (the denominator) to validate the sensitivity of detection. We recently showed that the perception of how often a variant is present in diagnostic specimens is greatly impacted by the depth of sampling, with frequency of detection of specific variants going up 10-fold in the specimens as the deep of sampling of the specimen increased (accepted manuscript).

## 10. Ideas on How to Design Validation Panels for Testing 1% Sensitivity

“I have to give a report on Romeo and Juliet tomorrow. I think the teacher expects me to read the whole play.”

There are several parameters that are of interest in validating a sequencing protocol. First, are true variants detected? Second, are false variants observed which would confuse the interpretation of true variants? Third, how accurately is the abundance of true variants reported? Fourth, how sensitive are mistakes to input viral template number?

In the recent WHO validation panel, the true abundance of minor variants was not known (since there is no validated method for their detection). Therefore to assess the various labs the “true” value was taken as the average value obtained by all of the labs [7]. Assuming there were differences in accuracy of the methods, such an approach would have the effect of making the less accurate results seem better and the more accurate results seem worse, probably not the ideal goal of a validation or testing panel. 

As an alternative approach we would suggest the following. First, a set of clinical specimens could be sequenced multiple times and at high depth (i.e., many TCS) with a UMI to provide accurate estimates of the composition of the population. Second, culture supernatants of 8E5 cells could be included as a negative control for false detection of variants. Third, multiple blinded aliquots of each specimen type could be tested at increasing dilution (limiting template number) to assess quality of template utilization. We would welcome a discussion of the strengths and weaknesses of designing NGS validation panels for the detection of minor variants using such an approach.

## Figures and Tables

**Figure 1 viruses-12-00850-f001:**
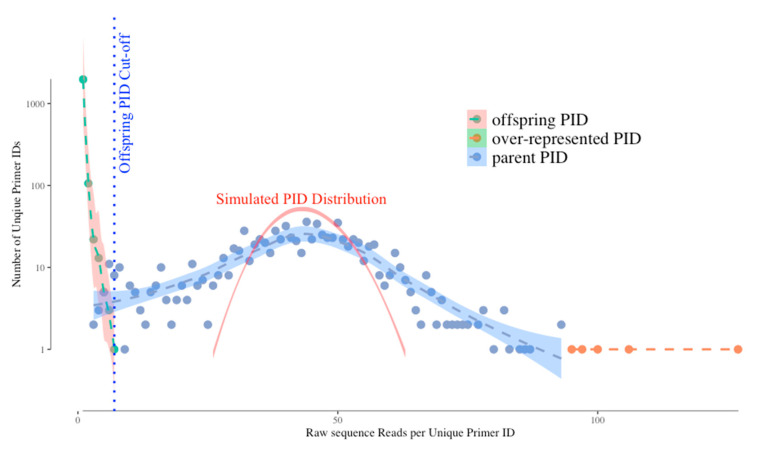
A Typical unique molecular identifier (UMI) (Primer ID) Frequency Distribution. The raw sequence output from a MiSeq run was sorted for individual UMIs and the number of reads for each UMI (after PCR) is plotted versus the number of UMIs with that read number. There is a large number of UMIs with very small read numbers (green dots to the left of the curve) representing offspring Primer IDs, i.e., Primer IDs generated from real Primer IDs by sequencing mistakes. The middle part of the graph shows the distribution of real Primer IDs (blue dots) and the fact that they are present in unequal numbers in the PCR product. This is not solely due to sampling of an otherwise equal distribution in the PCR product as a simulation of the same amount of the Primer IDs in this dataset (851) sampled 37,177 times (equal to the number of total raw reads in this dataset) to give the same mean read number as the observed distribution shows a much narrower spread (plotted as a red line in the figure). Finally, there are a small number of “jackpot” Primer IDs to the right (orange dots) that represent extreme PCR skewing and/or resampling of the Primer ID/UMI population. The blue dotted vertical line shows the offspring cut-off based on a simulation.

**Figure 2 viruses-12-00850-f002:**
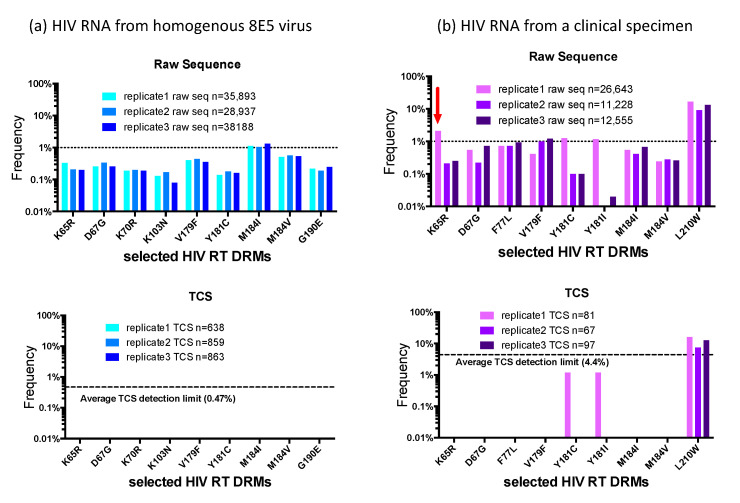
Comparison of Selected HIV-1 DRMs In the RT Region Analyzed Using Raw Sequence Reads or Template Consensus Sequences (TCS) Created Using Primer ID. (**a**) Sequencing of defective HIV-1 RNA from homogenous 8E5 virus. Top panel, analysis of DRMs in the raw sequence reads with the number of reads from triplicate sequencing runs shown (color coded); bottom panel is the same data analyzed using the Primer IDs in the data set to create TCS, with the number of TCS in each of the replicates shown. The dashed line in the top panel is an arbitrary 1% cutoff often used for reporting minor variants. The dashed line in the bottom panel is the average sampling depth (95% confidence of detection) for each of the triplicate sequencing runs. DRMs detected in the raw reads are shown. (**b**) Sequencing of HIV-1 RNA from a clinical specimen. These two panels are set up the same as in (**a**) with the lesser number of TCS in the data set indicated in the low panel and thus the reduced sensitivity of detection (dashed line).

**Figure 3 viruses-12-00850-f003:**
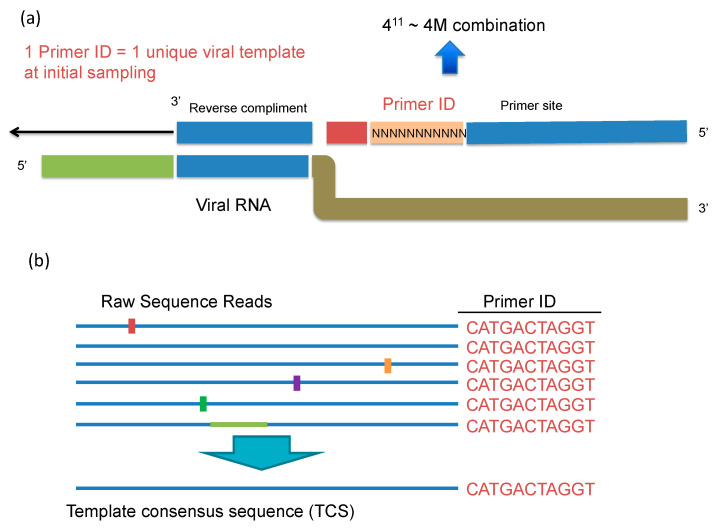
Illustration of Primer ID cDNA primer and how Primer ID reduces methods error (**a**) Primer ID cDNA primer. Ns in the Primer ID cDNA primer is a block of degenerate nucleotides. (**b**) Reads with same Primer ID will be collapsed to make a template consensus sequence based to greatly reduce errors. Colored vertical lines on the sequence reads indicate PCR mis-incorporations or sequencing errors, and the colored horizontal block indicates the PCR recombination.

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
