# Peer review of "Fact and Fiction about 1%: Next Generation Sequencing and the Detection of Minor Drug Resistant Variants in HIV-1 Populations with and without Unique Molecular Identifiers"

_viruses, 2020, doi:10.3390/v12080850_

Round 1

Reviewer 1 Report

Review of PGENETICS-D-19-01861: Measuring Selection across HIV Gag: Combining Review of Viruses-883568: Fact and Fiction About 1%: Next Generation Sequencing and the Detection of Minor Drug Resistant Variants In HIV-1 Populations With and Without Unique Molecular Identifiers

Authors: Shuntai Zhou and Ronald Swanstrom

This manuscript is a review and discussion of the "Primer ID" technology that has been extensively developed by the authors previously and is used with PCR/NGS to help overcome sampling errors when quantifying variants in viral populations within a host. The authors elaborate the statistical advantages of the method and give numerical illustrations, and give intuition and estimates for the sampling error rates of PCS/NGS with and without Primer ID.

I found the illustration of the advantages of Primer ID to be clear and well laid out, and this manuscript would be a useful addition to the literature on Primer ID.

Comments:

The authors do not give many details here about how offspring PIDs and over-represented PIDs are distinguished from parent PIDs, and it seems like this could contribute to sampling error, the subject of the review. In figure 1 there is no sharp break in the curve between the types. Is it sometimes ambiguous whether a read is a parent or offspring? Does this affect the sampling error analysis? Would using a longer ID sequence make the offspring more obvious? It might be worth adding some more description of the criteria, while still leaving the fine details to citations.

Around line 449 the authors say 50 templates are needed to improve accuracy in abundance measurements. Presumably, 50 are needed to get a certain % confidence interval. To clarify where "50" comes from, perhaps mention what this interval is.

Also, the derivation of the equation 300/N isn't given. This might be all right. For a confidence of detection c, the general approximation is p ~ -log(1-c)/N.

Template switching and recombination during PCR can result in errors in mutation frequencies and cause spurious apparent correlations between different sites posing difficulties in scaling up of assays as well as correlation-based studies of protein sequence, structure and function. The authors should comment in more detail on how template switching or recombination during NGS can affect both the accuracy and precision of reported mutation frequencies, especially for those of minority variants, affecting the confidence interval as well, and how using a UMI based protocol can avoid these spurious effects  introduced by recombination.

It is not clear to me how the authors' multiplex protocol using 4 or 5 cDNA primers with ID in a single cDNA reaction helps to overcome read length limitations of the MiSeq platform (line 418-427). Given the sequence read length limitations of most sequencing platforms, the inclusion of a primer can further reduce coverage for the genomic region of interest and contribute to the loss of linkage. The authors should explain this in greater detail.

The manuscript discusses at length the issues introduced as a result of PCR into NGS and pitfalls therein, and the gold standard sequencing method for HIV drug resistance clinical testing remains Sanger sequencing. The authors should perhaps also mention how errors or artifacts due to PCR can also come up in Sanger sequencing and how best to overcome or address them.

Other Comments:

Is there a reason the authors change the terminology from "Primer ID" to "UMI"? "Primer ID" seems to be in common use by these and other authors now. Even in the current manuscript, Primer ID or PID is used in some places and UMI in others, or sometimes the combined "Primer
 ID/UMI". If the authors want to try to change terminology going forward, this review article could be a good place to give an argument why.

I would also be curious for citations related to incorrect claims of 1% sensitivity using NGS. The authors say it is very common (line 159), but no citation.

Typos:
Line 345: Fig 2
Line 468: depth

Author Response

Response to Reviewer 1

This manuscript is a review and discussion of the "Primer ID" technology that has been extensively developed by the authors previously and is used with PCR/NGS to help overcome sampling errors when quantifying variants in viral populations within a host. The authors elaborate the statistical advantages of the method and give numerical illustrations, and give intuition and estimates for the sampling error rates of PCS/NGS with and without Primer ID.

I found the illustration of the advantages of Primer ID to be clear and well laid out, and this manuscript would be a useful addition to the literature on Primer ID.

We thank the reviewer for the careful and thoughtful review of the article. The comments and suggestions are very helpful. We appreciate the reviewer’s efforts in helping us improve our manuscript.

Comments:

The authors do not give many details here about how offspring PIDs and over-represented PIDs are distinguished from parent PIDs, and it seems like this could contribute to sampling error, the subject of the review. In figure 1 there is no sharp break in the curve between the types. Is it sometimes ambiguous whether a read is a parent or offspring? Does this affect the sampling error analysis? Would using a longer ID sequence make the offspring more obvious? It might be worth adding some more description of the criteria, while still leaving the fine details to citations.

The reviewer raised a very important question. We agree with the reviewer that there is hardly any sharp break in the curve in most of cases. It is true that the offspring can potentially affect the sampling quality to some degree (but not much because most of the offspring PIDs only appear a few times). In our previous paper (Zhou, et al, JVI, 2015) we discussed the details of the origin of offspring PIDs. We did simulations in that paper to make a frequency cut-off based on two factors: 1) the frequency of the most abundant PID (and in later versions of the TCS pipeline, we used an average number of the top 5 most abundant PID), and the MiSeq raw sequencing error profile (adjustable in later version of TCS pipeline). Using these two factors we ran a simulation to see under these conditions what is the highest frequency of offspring PIDs can we see, and then we make a cut-off based on that value. We may lose some true PIDs using this cut-off model because in some cases the offspring distribution curve is overlapping with distribution of the actual PIDs.

Using a longer PID actually does not affect the offspring very much. We ran simulations using PID length from 8 to 12 and the cut-off values were very similar.

We modified figure 1 to show the simulated offspring PIDs distribution with the parent PID and over-represented PIDs, as well as the offspring cut-off. We included more explanation of the offspring PIDs in the manuscript.

Around line 449 the authors say 50 templates are needed to improve accuracy in abundance measurements. Presumably, 50 are needed to get a certain % confidence interval. To clarify where "50" comes from, perhaps mention what this interval is.

We thank the authors for the comment. We used 50 as an example number for increasing the sensitivity of detection and for improving the accuracy of estimating abundance of a variant present at 20%. This is a hypothetical discussion but based on real world starting numbers. Because the discussion involves both sensitivity and accuracy, we have added text to try to clarify what is accomplished as a function of sampling depth.

Also, the derivation of the equation 300/N isn't given. This might be all right. For a confidence of detection c, the general approximation is p ~ -log(1-c)/N.

We appreciated this comment. We used 300/N as a rough estimation of the exact binomial distribution. The origin of this short cut formula is lost to history as one of the authors picked it up from some forgotten statistician long ago as a quick way to make an estimation. However, in comparisons with the exact binomial formula it is surprisingly robust. We believe it is helpful for people who are not conversant with statistics as a quick way to see what their sensitivity of detection is with a very simple calculation. Also, since our audience is a group of people who have only ever made assumptions about their depth of sampling, giving them the simplest of equations to get them thinking what their data might really be telling them should be helpful. We do point out in the manuscript that the exact binomial test should be used to calculate a more precise detection limit while 300/N is a rough estimation.

Template switching and recombination during PCR can result in errors in mutation frequencies and cause spurious apparent correlations between different sites posing difficulties in scaling up of assays as well as correlation-based studies of protein sequence, structure and function. The authors should comment in more detail on how template switching or recombination during NGS can affect both the accuracy and precision of reported mutation frequencies, especially for those of minority variants, affecting the confidence interval as well, and how using a UMI based protocol can avoid these spurious effects  introduced by recombination.

We agree with the reviewer that the template switching and recombination of PCR is also a source of errors. We added a discussion on template switching and recombination during PCR in our manuscript (around line 410).

It is not clear to me how the authors' multiplex protocol using 4 or 5 cDNA primers with ID in a single cDNA reaction helps to overcome read length limitations of the MiSeq platform (line 418-427). Given the sequence read length limitations of most sequencing platforms, the inclusion of a primer can further reduce coverage for the genomic region of interest and contribute to the loss of linkage. The authors should explain this in greater detail.

As the reviewer implies, the inclusion of multiple primers does not change the limitation in the read length of the MiSeq instrument. What our multiplex approach does is allow a user to query multiple regions of the viral genome in a single reaction. There is linkage within each individual amplicons but no linkage between the amplicons. We expanded the discussion of this approach with an example of how we set up such a reaction and the typical output (in the long paragraph starting around line 438). We have also included a citation to a paper just published by our group using this protocol. We do have a more detailed manuscript in preparation that will provide complete documentation of this approach but we want people to appreciate that it is available and provides the information about depth of sequencing while also providing information about minor variants over a larger portion of the genome.

The manuscript discusses at length the issues introduced as a result of PCR into NGS and pitfalls therein, and the gold standard sequencing method for HIV drug resistance clinical testing remains Sanger sequencing. The authors should perhaps also mention how errors or artifacts due to PCR can also come up in Sanger sequencing and how best to overcome or address them.

This is not the focus of our paper. Sanger sequencing is quite robust for generating a consensus sequence. We don't have experience in troubleshooting issues that come up with Sanger sequencing and therefore are the wrong people to address this question.

Other Comments:

Is there a reason the authors change the terminology from "Primer ID" to "UMI"? "Primer ID" seems to be in common use by these and other authors now. Even in the current manuscript, Primer ID or PID is used in some places and UMI in others, or sometimes the combined "Primer

 ID/UMI". If the authors want to try to change terminology going forward, this review article could be a good place to give an argument why.

We thank the reviewer for this comment. While we were developing the “Primer ID” approach, other groups who sequenced DNA developed the molecular tagging in parallel, in which they called it “UMI”, and the term of UMI seems to be adopted quite widely by DNA NGS. That is the reason why we used UMI in this review, as it is a broader term for this type of technology. Since we developed this approach as “Primer ID” we like to use that term but we have tried to make the point that it is a version of the “UMI” strategy. 

I would also be curious for citations related to incorrect claims of 1% sensitivity using NGS. The authors say it is very common (line 159), but no citation.

We thank the reviewer for the comment. We include a recently published article directly comparing 10 different NGS HIVDR assays, in which most of them used 1% as the minority cut-off. We also cited a HIVDR NGS assay pipeline comparison paper published in Scientific Report early this year, in which the authors claimed that the 5 pipelines used “a median frequency threshold of ≥1%” for minority variants.

Typos:

Line 345: Fig 2

Line 468: depth

We have corrected the typos.

Reviewer 2 Report

This is an excellent paper from Zhou and Swanstrom that discusses technical issues related to deep-sequencing and the sensitivity thereof. First, the article is incredibly well written, and this reviewer enjoyed reading it. Second, there are many misconceptions about NGS sensitivity for detecting minor variants in viral populations. The authors clearly highlight the limitations of may approaches and strongly justify the need for Primer ID technology. Third, this is an important area of research and this perspective provides an excellent overview about to how understand and experimentally approach the problem. Excellent paper!

Author Response

Response to Reviewer 2

This is an excellent paper from Zhou and Swanstrom that discusses technical issues related to deep-sequencing and the sensitivity thereof. First, the article is incredibly well written, and this reviewer enjoyed reading it. Second, there are many misconceptions about NGS sensitivity for detecting minor variants in viral populations. The authors clearly highlight the limitations of may approaches and strongly justify the need for Primer ID technology. Third, this is an important area of research and this perspective provides an excellent overview about to how understand and experimentally approach the problem. Excellent paper!

We thank the reviewer for these kind comments about our manuscript. No issues were raised.